# Chronic lithium treatment elicits its antimanic effects via BDNF-TrkB dependent synaptic downscaling

Erinn S Gideons[†], Pei-Yi Lin, Melissa Mahgoub, Ege T Kavalali, Lisa M Monteggia*

Department of Neuroscience, UT Southwestern Medical Center, Dallas, United States

**Abstract** Lithium is widely used as a treatment for Bipolar Disorder although the molecular mechanisms that underlie its therapeutic effects are under debate. In this study, we show brain-derived neurotrophic factor (BDNF) is required for the antimanic-like effects of lithium but not the antidepressant-like effects in mice. We performed whole cell patch clamp recordings of hippocampal neurons to determine the impact of lithium on synaptic transmission that may underlie the behavioral effects. Lithium produced a significant decrease in α-amino-3-hydroxyl-5-methyl-4-isoxazolepropionic acid receptor (AMPAR)-mediated miniature excitatory postsynaptic current (mEPSC) amplitudes due to postsynaptic homeostatic plasticity that was dependent on BDNF and its receptor tropomyosin receptor kinase B (TrkB). The decrease in AMPAR function was due to reduced surface expression of GluA1 subunits through dynamin-dependent endocytosis. Collectively, these findings demonstrate a requirement for BDNF in the antimanic action of lithium and identify enhanced dynamin-dependent endocytosis of AMPARs as a potential mechanism underlying the therapeutic effects of lithium.

*For correspondence: lisa.
monteggia@utsouthwestern.edu

Present address: [†]Department
of Molecular Neurobiology, Max
Planck Institute for Experimental
Medicine, Goettingen, Germany

Competing interests: The
authors declare that no
competing interests exist.

Reviewing editor: Inna Slutsky,
Tel Aviv University, Israel

## Introduction

Lithium was initially reported for treatment as a mood stabilizer over 60 years ago (*Cade, 1949*) and is still widely used for the treatment of Bipolar Disorder (*Mitchell, 2013*; *Poolsup et al., 2000*; *Vieta and Valentí, 2013*). However, despite its extensive use for the treatment of bipolar disorder, the cellular and molecular mechanisms underlying its antimanic and antidepressant responses remain poorly understood (*Can et al., 2014*; *Malhi and Outhred, 2016*). Putative mechanisms for the thera-peutic effects of lithium include inhibition of glycogen synthase kinase-3β (GSK3β) (*Klein and Mel-ton, 1996*), upregulation of neurotrophins, as well as their receptors, and downregulation of α-amino-3-hydroxy-5-methyl-4-isoxazolepropionic acid receptor (AMPAR) expression (*Ankolekar and Sikdar, 2015*; *Du et al., 2004*, *2010*; *Gray et al., 2003*; *Seelan et al., 2008*; *Wei et al., 2010*) among others. However, which —if any— of these effects of lithium are the primary mechanism for treatment for bipolar disorder is currently unknown.

Earlier studies suggested a potential link between the action of lithium as a mood stabilizer and neurotrophins, in particular brain-derived neurotrophic factor (BDNF). BDNF and aberrant signaling through its high affinity receptor tropomyosin receptor kinase B (TrkB) have been proposed to underlie both the pathophysiology and treatment of bipolar disorder (*Autry and Monteggia, 2012*; *Malhi et al., 2013*; *Scola and Andreazza, 2015*). Previous work has shown that patients with bipolar disorder have decreased peripheral BDNF mRNA in blood lymphocytes and monocytes in compari-son to healthy controls (*D'Addario et al., 2012*). Moreover, both manic and depressive states in patients with bipolar disorder have been associated with significantly decreased BDNF blood serum levels compared to patients in euthymic states and healthy controls (*Fernandes et al., 2015*;

**eLife digest** Nerve cells, or neurons, communicate with each other by releasing chemical messengers that bind to and activate receptor proteins on the surface of the other cells. The chemicals affect the connections between neurons, and many diseases – including bipolar disorder – are related to there being too much or too little of these chemicals in the brain. Patients with bipolar disorder experience periods of both depression and mania. During a manic episode, affected individuals typically feel elated and have more energy than usual despite needing less sleep, but also can also be irritable and impulsive.

The exact cause of bipolar disorder is unknown. Patients with bipolar disorder often have low levels of a protein called brain-derived neurotrophic factor, or BDNF for short, which plays an essential role in keeping the brain healthy, and may also regulate the connections between neurons. One of the main treatments for bipolar disorder, a mood stabilizer called lithium, has also been linked to BDNF in previous studies; however, the details of the interaction were not clear.

Gideons et al. studied how lithium works by feeding mice food pellets that contained lithium. After a few weeks, the mice had concentrations of lithium in their blood comparable to those of people taking the drug, as well as increased levels of BDNF in the brain. Gideons et al. then examined if BDNF was needed for the lithium's ability to treat manic episodes. Mice exposed to another drug, amphetamine, normally move around a lot, mimicking the increased energy of someone with mania. As expected, feeding normal mice lithium blocked this effect of amphetamine, but feeding lithium to mutant mice that lack BDNF did not. This indicates that BDNF is indeed needed for the antimanic effect of lithium. Further experiments showed that BDNF is not needed for lithium's antidepressant effect.

By studying the animals' brains, Gideons et al. went on to show that the lithium-fed mice had weaker connections between their neurons than mice that had eaten standard food. In the lithium-fed mice, many of the receptor proteins had been reabsorbed back into the neurons, lowering the ability of neurons to communicate with one another. This process depended on BDNF, suggesting that this protein is essential for lithium to suppress the connections between neurons.

Taken together, these results reveal that the effects of lithium on both an animal's brain and its behavior rely on BDNF. This knowledge should make it easier to develop new strategies and identifying new molecularly specific targets for treating bipolar disorder as well as other neuropsychiatric diseases.

Tunca et al., 2014). Post-mortem analysis of hippocampal tissue has also revealed reduced BDNF protein levels in patients with bipolar disorder (Knable et al., 2004). In rodents, experimental interventions that cause manic-like (Frey et al., 2006; Fries et al., 2015; Jornada et al., 2010) and depressive-like behaviors (Smith et al., 1995; Tsankova et al., 2006; Ueyama et al., 1997) have also been shown to result in decreased BDNF mRNA and protein levels in the hippocampus. In contrast, lithium treatment is associated with increased BDNF protein levels in the serum of patients with bipolar disorder (Cunha et al., 2006; de Sousa et al., 2011; Tramontina et al., 2009). In rodent models, chronic lithium treatment has been shown to increase BDNF mRNA and protein expression in hippocampus, cortex, and amygdala as well as cortical neurons in culture (Fukumoto et al., 2001; Jornada et al., 2010; Yasuda et al., 2009). Lithium treatment has also been shown to increase TrkB activity in neuronal cultures suggesting an increase in BDNF-TrkB signaling (Hashimoto et al., 2002).

While previous work has demonstrated that BDNF is necessary for the antidepressant response of conventional antidepressants (Adachi et al., 2008; Ibarguen-Vargas et al., 2009; Monteggia et al., 2004, 2007), as well as the rapid antidepressant effects of ketamine (Autry et al., 2011; Lepack et al., 2014), it is currently unknown whether BDNF or the TrkB receptor are required for lithium's antidepressant or antimanic effects. In this study, we establish the necessity of BDNF-TrkB signaling in the antimanic-like but not antidepressant-like response to lithium. We also identify a direct effect of lithium treatment on AMPAR trafficking, where AMPA receptor surface expression is decreased due to a sustained increase in dynamin mediated AMPAR endocytosis. This process is

dependent on BDNF-TrkB function and results in synaptic downscaling where unitary postsynaptic responses are decreased in a manner proportional to their relative strengths. Collectively, our findings reveal a requirement for neurotrophic signaling in the behavioral and cellular effects of lithium. These data provide novel mechanistic insight into the action of lithium which may underlie its therapeutic effect for the treatment of bipolar disorder.

## Results

### Chronic lithium treatment increases BDNF expression

Previous studies have used mice to examine the impact of lithium treatment on molecular signaling. However, typically the lithium serum concentration is not determined in mice following lithium treatment leaving it unclear whether reported changes occur within the therapeutic range. We established a lithium protocol in which C57BL/6 mice were given 0.2% lithium chloride (LiCl) chow for four days, followed by 0.4% LiCl chow for the remainder of the treatment and testing period, which lasted a total of 11–17 days depending on the experiment (*Figure 1A*). This protocol resulted in a lithium serum concentration in treated mice that was within the therapeutic range of 0.5–2 mM (*Amdisen, 1980*), with an average concentration of 1 mM (*Figure 1B*). Control (CTL) mice were given the same chow without lithium. Following 11 days of lithium treatment, mice were sacrificed and the hippocampus rapidly removed. To confirm our lithium protocol produced the expected molecular effects, we examined GSK3 inhibition (*Klein and Melton, 1996*; *Stambolic et al., 1996*), by measuring the phosphorylation of GSK3$\beta$ at serine nine with western blot analysis and found a significant increase relative to total GSK3$\beta$ (*Figure 1C*). We next examined whether the lithium treatment regulated the expression of BDNF in the hippocampus. Using Q-PCR targeted to the coding exon of *Bdnf*, we found lithium treated mice had a significant increase in *Bdnf* mRNA expression (*Figure 1D*). We also found a significant increase in BDNF protein expression (*Figure 1E*) in lithium treated mice relative to CTL mice.

### BDNF is required for the antimanic-like effect of lithium

Since lithium increases the expression of BDNF mRNA and protein levels, we investigated whether BDNF was necessary for lithium's behavioral action. Inducible *Bdnf* KO mice, in which *Bdnf* is deleted ~70–80% in broad forebrain regions (*Monteggia et al., 2004*), were given lithium treated chow following our protocol (*Figure 1A*). To determine whether this lithium treatment produced the expected behavioral effects, we tested mice in the forced swim test (FST), a test with predictive value for antidepressant responses (*Porsolt et al., 1977*). Littermate control (CTL) mice treated with lithium showed a significant reduction in immobility time that was suggestive of an antidepressant-like response compared to CTL mice that did not receive lithium (*Figure 2A*). In agreement with previous data we found that loss of *Bdnf* in the inducible *Bdnf* KO mice did not alter immobility in the FST compared to CTL mice (*Monteggia et al., 2004*). We also found that *Bdnf* KO mice administered lithium showed a significant reduction in immobility comparable to KO mice that did not receive lithium (*Figure 2A*) suggesting that BDNF is not necessary for the antidepressant-like effect of lithium in the FST.

To investigate whether BDNF was required for lithium's antimanic-like effects, we used the amphetamine hyperlocomotor test in which lithium blunts the increased locomotor activity that occurs following acute amphetamine injection, a commonly used assay for the antimanic-like effects of lithium in mice (*Flaisher-Grinberg and Einat, 2010*; *Gould et al., 2007*). The amphetamine induced hyperlocomotor test was performed subsequent to the FST in the same cohorts of mice. Littermate CTL mice treated with acute amphetamine have the expected significant increase in the total number of horizontal beam breaks compared to vehicle treated mice (*Figure 2B*). CTL mice receiving lithium treatment showed a small non-significant increase in the total number of beam breaks compared to vehicle treated CTL mice, and were indistinguishable from chronic lithium treated mice receiving acute amphetamine, indicating that lithium treatment could prevent the hyperactivity induced by amphetamine (*Figure 2B*). In contrast, while inducible *Bdnf* KO mice receiving acute amphetamine showed a significant increase in the number of beam breaks compared to vehicle treated *Bdnf* KOs, and a slight increase with chronic lithium treatment, the chronic lithium treated *Bdnf* KO mice receiving acute amphetamine still had a significant increase in the number of

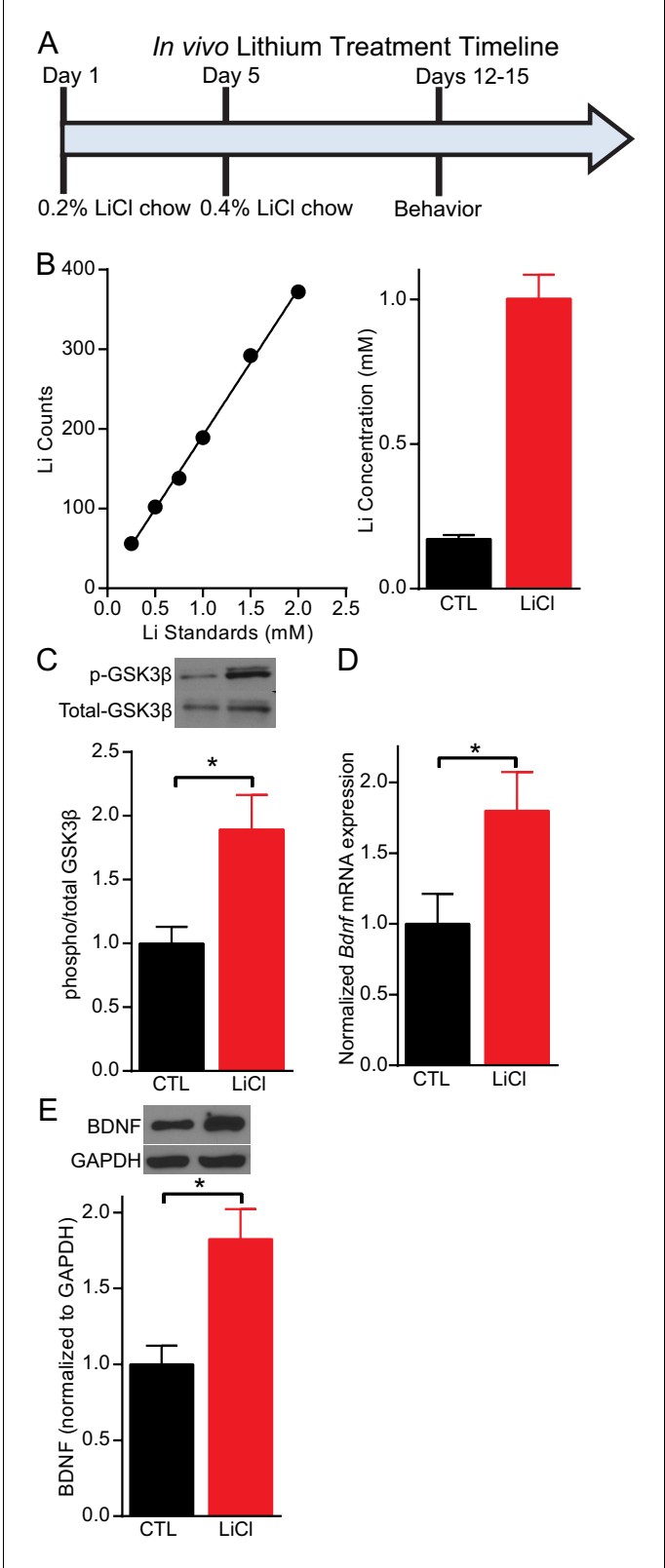

**Figure 1.** Lithium serum concentration within the therapeutic range increases BDNF expression in mice. (**A**) Timeline of chronic lithium chloride (LiCl) treatment and behavioral testing. (**B**) (Left) Example standard curve of lithium counts used to calculate lithium concentration in blood serum. (Right). Average lithium concentration in blood serum in control and treated mice (*n*= 9–12 per group). (**C**) Chronic lithium treatment in mice produces a

*Figure 1 continued on next page*

*Figure 1 continued*

significant decrease in immobility in the FST compared to the control group (Student's unpaired *t* test *p<0.0001, *n* = 10 mice per group). (D) Chronic lithium treatment caused a significant increase in *Bdnf* mRNA expression in the hippocampus (Student's unpaired *t* test *p=0.04, *n* = 8–9 mice per group). (E) Chronic lithium treatment caused a significant increase in BDNF protein expression in the hippocampus (Student's unpaired *t* test *p=0.002, *n* = 10 mice per group).

beam breaks comparable to amphetamine treatment alone. These data demonstrate that lithium treatment does not block the hyperlocomotor effects of amphetamine in inducible *Bdnf* KO mice and thus suggests that BDNF is required for the antimanic-like effect of lithium.

We next assessed whether the loss of BDNF impacted lithium's effect on synaptic efficacy from hippocampal slices. Inducible *Bdnf* KO and littermate control mice were given lithium treated chow or untreated chow (*Figure 1A*). The slope of the input/output (I/O) curve, which was plotted as the presynaptic volley versus the fEPSC slope, was taken as a measure of synaptic efficacy associated with synaptic density or strength of individual synapse. We found that in CTL mice, chronic lithium treatment resulted in a significant reduction in the slope of the I/O curve in hippocampal slices in comparison to mice not treated with lithium (*Figure 2C*), indicating that chronic lithium treatment reduces hippocampal synaptic strength. Inducible *Bdnf* KO mice that did not receive lithium had a similar slope of the I/O curve compared to untreated littermate CTL mice (*Figure 2C*) showing that the loss of BDNF did not impact synaptic efficacy. We also found that the slope of the I/O curve in the hippocampal slices from the inducible *Bdnf* KO mice treated with lithium was also unchanged in comparison to the untreated CTL and *Bdnf* KO mice (*Figure 2C*), suggesting that BDNF expression is required for the lithium-mediated reduction of synaptic strength.

## BDNF and TrkB are required for lithium-mediated decreases in AMPAR mEPSC amplitudes

Previous work has shown that lithium treatment produces a significant decrease in AMPAR miniature excitatory postsynaptic current (mEPSC) amplitudes in cultured neurons, which has been suggested to underlie the antimanic effects of lithium (*Ankolekar and Sikdar, 2015*; *Du et al., 2008*; *Wei et al., 2010*). We therefore incubated dissociated C57BL/6 hippocampal neurons with 1 mM LiCl or 1 mM NaCl, to control for changes in osmolarity, for 11–15 days and then recorded AMPAR mEPSCs (*Figure 3A*). Lithium treatment resulted in a significant decrease in AMPAR mEPSC amplitudes in comparison to NaCl treated or untreated CTL neurons (*Figure 3B–D*). To examine whether lithium's effect on AMPAR mEPSC amplitudes was due to synaptic scaling, we plotted in rank order the amplitudes from each condition with a linear fit equation. We found that lithium treatment resulted in a 41% and 26% decrease in the slope compared to untreated and NaCl treatment, respectively (*Figure 3E*). These data show lithium produces a downward scaling of all AMPAR mEPSC amplitudes and impacts post-synaptic homeostatic plasticity. Notably, lithium did not produce generalized effects on synaptic measures as there were no changes in mEPSC frequency compared to untreated or NaCl treated neurons (*Figure 3F*). Since lithium's effects on AMPAR mEPSC amplitudes were due to synaptic scaling, which can be influenced by AMPAR trafficking, we examined the surface expression of the AMPAR subunit, GluA1. We performed surface biotinylation experiments and found lithium treatment results in a significant decrease in GluA1 surface expression relative to total GluA1 in comparison to untreated or NaCl treated neurons (*Figure 3G*).

Since our data demonstrated that BDNF was required for lithium's antimanic-like effects in mice and lithium-mediated decrease in synaptic strength, we examined whether chronic lithium treatment's effects on synaptic function, which may ultimately trigger the behavioral effects, were dependent on BDNF and its high affinity receptor TrkB. BDNF is a secreted protein that has been shown to influence synaptic upscaling and downscaling, depending on the brain region examined (*Leslie et al., 2001*; *Reimers et al., 2014*; *Rutherford et al., 1998*). We therefore examined whether lithium's effects on AMPAR mEPSC amplitudes were dependent on BDNF. We cultured hippocampal neurons from *Bdnf* ^fl/fl^ mice and infected them with lentivirus expressing Cre recombinase (Cre) tagged with GFP to delete the gene of interest or GFP alone as a control. Previous studies from our laboratory have shown that this lentivirus construct expressing Cre tagged with GFP can efficiently

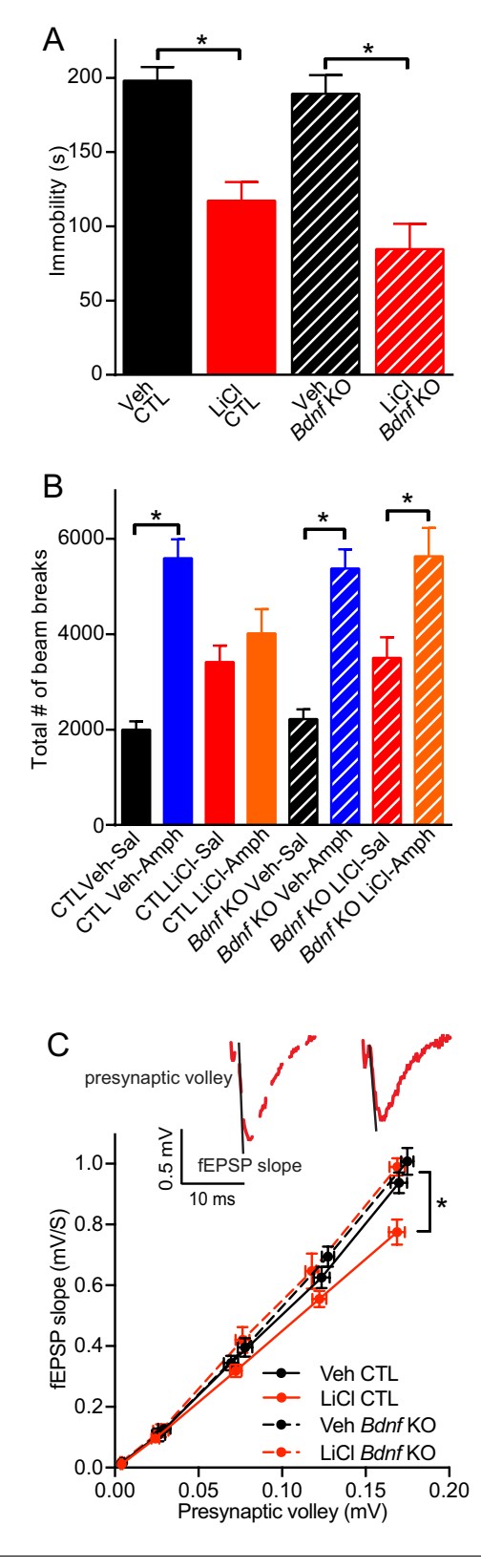

**Figure 2.** BDNF expression is required for the antimanic-like effect of lithium. (**A**) Chronic lithium treatment caused a significant decrease in the *Figure 2 continued on next page*

knock out endogenous gene expression without triggering cell death (*Akhtar et al., 2009*; *Nelson et al., 2006*). As expected, GFP infected neurons treated with chronic lithium had a significant decrease in AMPAR mEPSC amplitudes in comparison to untreated or NaCl treated neurons (*Figure 4A–D*). In BDNF deficient cultures, *Bdnf^fl/fl* neurons infected with Cre, there was no difference in AMPAR mEPSC amplitudes compared to GFP infected neurons suggesting loss of BDNF does not impact this synaptic measure (*Figure 4A–D*). In contrast to wild-type neurons, we found chronic lithium treatment of BDNF deficient cultures induced a small but nonsignificant decrease in AMPAR mEPSC amplitudes (*Figure 4D*) suggesting a requirement for BDNF. We also examined mEPSC frequency from these six treatment groups and did not find a significant change with any condition (data not shown). Taken together, these data suggest that chronic lithium's effect on AMPAR mEPSC amplitudes is dependent on BDNF.

To further evaluate whether BDNF signaling is required for lithium's effects on AMPAR mediated synaptic transmission, we explored the requirement for the TrkB receptor. We cultured hippocampal neurons from *Ntrk2^fl/fl* mice and infected them with either lentivirus expressing Cre or GFP. Consistent with our previous results, GFP infected neurons treated with lithium had a significant decrease in AMPAR mEPSC amplitude compared to untreated or NaCl treated neurons (*Figure 5A–D*). We found *Ntrk2^fl/fl* neurons infected with CreGFP, had indistinguishable AMPAR mEPSC amplitudes compared to GFP infected neurons demonstrating that loss of TrkB does not affect this synaptic measure (*Figure 5D*). In agreement with data from the *Bdnf* deficient neurons, we found lithium treatment of *Ntrk2* deficient neurons did not affect AMPAR mEPSC amplitudes showing a requirement for TrkB in lithium's effect on AMPAR mEPSC responses (*Figure 5D*). We measured mEPSC frequency in the six treatment groups and did not observe any significant differences demonstrating a specific effect of lithium on mEPSC properties (data not shown).

## Dynamin-dependent endocytosis is required for lithium-mediated decrease in AMPAR mEPSC amplitudes

Our data so far shows that chronic lithium treatment of hippocampal neurons leads to a significant decrease in AMPAR mEPSC amplitudes that is dependent on BDNF-TrkB signaling. The

*Figure 2 continued*

immobility time in the FST in littermate CTL and inducible *Bdnf* KO mice in comparison to vehicle chow treated mice (ANOVA $F_{3,90}$ = 18.1 *p<0.0001, Dunnett's multiple comparisons Veh CTL vs LiCl CTL *p<0.0001, Veh *Bdnf* KO vs LiCl *Bdnf* KO *p<0.0001, n = 19–28 mice per group). (B) Acute amphetamine injection caused a significant increase in locomotor activity in control and *Bdnf* KO mice in comparison to the saline injected mice. Lithium treatment resulted in a nonsignificant increase in locomotor activity of CTL and *Bdnf* KO mice compared to vehicle treated mice. Chronic lithium treatment blunted the increased locomotor activity following acute amphetamine injection in the CTL littermate mice, but not in the *Bdnf* KO mice (ANOVA $F_{7,98}$ = 12.75 p<0.0001, Tukey's multiple comparisons CTL Veh-Sal vs CTL Veh-Amph *p<0.0001, CTL LiCl-Sal vs CTL LiCl-Amph p=0.97, CTL Veh-Sal vs CTL LiCl-Sal p=0.23, *Bdnf* KO Veh-Sal vs *Bdnf* KO Veh-Amph *p<0.0001, *Bdnf* KO LiCl-Sal vs *Bdnf* KO LiCl-Amph *p=0.01, *Bdnf* KO Veh-Sal vs *Bdnf* KO LiCl-Sal p=0.44, n = 11–15 mice per group). (C) Chronic lithium treatment resulted in a 20% reduction in the I/O curve in the CTL mice chronically treated with LiCl chow in comparison to Veh CTL mice, which is plotted as the slope of the fEPSP is plotted as a function of the presynaptic fiber volley. Linear fit slopes were calculated for Veh CTL(5.53 ± 0.2) vs CTL LiCl (4.656 ± 0.03825). There were no significant differences in the fEPSP slope between Veh *Bdnf* KO slices (5.85 ± 0.178) and LiCl treated *Bdnf* KO (5.899 ± 0.1703). (Two-way ANOVA Interaction $F_{1,29}$ = 3.24 p=0.08, Row (Genotype) $F_{1,29}$ = 5.09 *p=0.032, Column (Diet) $F_{1,29}$ = 12.61 *p=0.001. Sidak's multiple comparisons Veh CTL vs LiCl CTL *p=0.029, Veh *Bdnf* KO vs LiCl *Bdnf* KO p=0.999, Veh CTL vs Veh *Bdnf* KO p=0.739, LiCl CTL vs LiCl *Bdnf* KO *p=0.007 n = 6–10 recordings per group).

decrease in AMPAR mEPSC amplitudes and the reduced GluA1 surface membrane expression suggests lithium is augmenting AMPAR endocytosis. Therefore, we examined whether inhibiting AMPAR endocytosis would rescue the synaptic phenotype observed with lithium treatment. AMPAR endocytosis is dependent on the GTPase dynamin (*Lu et al., 2007*), which is expressed by three related genes in mammals (*Raimondi et al., 2011*). Dynamin 1 and 3 (*Dnm1* and *Dnm3*) are highly expressed in neurons and found in dendrites (*Calabrese and Halpain, 2015*; *Noda et al., 1993*). Previous work has shown that constitutive $Dnm1^{-/-}$ / $Dnm3^{-/-}$ double KO mice are not viable (*Raimondi et al., 2011*). Therefore, to examine the contribution of both dynamin1 and dynamin3 on lithium mediated synaptic effects, we crossed $Dnm1^{fl/fl}$ mice with $Dnm3^{-/-}$ mice as the constitutive deletion of *Dnm3* does not impact synaptic function (*Raimondi et al., 2011*). We cultured $Dnm1^{fl/fl}$ / $Dnm3^{-/-}$ neurons and infected them with lentivirus expressing Cre to delete *Dnm1* and thus generate the double KOs, or GFP as a control. The GFP infected neurons treated with lithium had a significant reduction in AMPAR mEPSC amplitudes compared to untreated neurons showing that the loss of *Dnm3* does not impact the effect of lithium on synaptic function (*Figure 6A–C*). In contrast, deletion of both *Dnm1* and *Dnm3* occluded the effect of chronic lithium treatment on AMPAR mEPSC amplitudes in comparison to untreated cultures demonstrating that lithium is potentiating dynamin dependent endocytosis of AMPAR (*Figure 6C*).

To control for possible compensatory effects due to the loss of *Dnm1* and *Dnm3*, in our next experiments we cultured hippocampal neurons from C57BL/6 mice and acutely inhibited dynamin activity in the postsynaptic neuron. We chronically treated neurons with lithium and then added the pan-dynamin inhibitor Dyngo (*McCluskey et al., 2013*), or the vehicle 1% DMSO, to the internal pipette solution prior to our recordings. Consistent with previous data, chronic lithium treatment resulted in a significant decrease in AMPAR mEPSC amplitudes in neurons recorded with vehicle in the pipette solution (*Figure 6D–F*). In contrast, the presence of Dyngo in the pipette solution rescued the decrease in AMPAR mEPSC amplitudes seen after lithium treatment (*Figure 6F*). Taken together, these experiments demonstrate that chronic lithium treatment enhances AMPAR endocytosis to alter synaptic function in a dynamin dependent manner and the loss of dynamin, specifically *Dnm1* and *Dnm3*, can rescue the synaptic deficits.

## Discussion

In the current study, we report that lithium requires BDNF-TrkB signaling to mediate its antimanic as well as synaptic effects. Chronic lithium treatment in vivo results in a sustained increase in BDNF mRNA and protein expression. However, while BDNF is required for the antimanic effects of lithium it is not required for the antidepressant effects. We determined that chronic lithium treatment in vivo caused a significant decrease in I/O curves from hippocampal slices that were dependent on

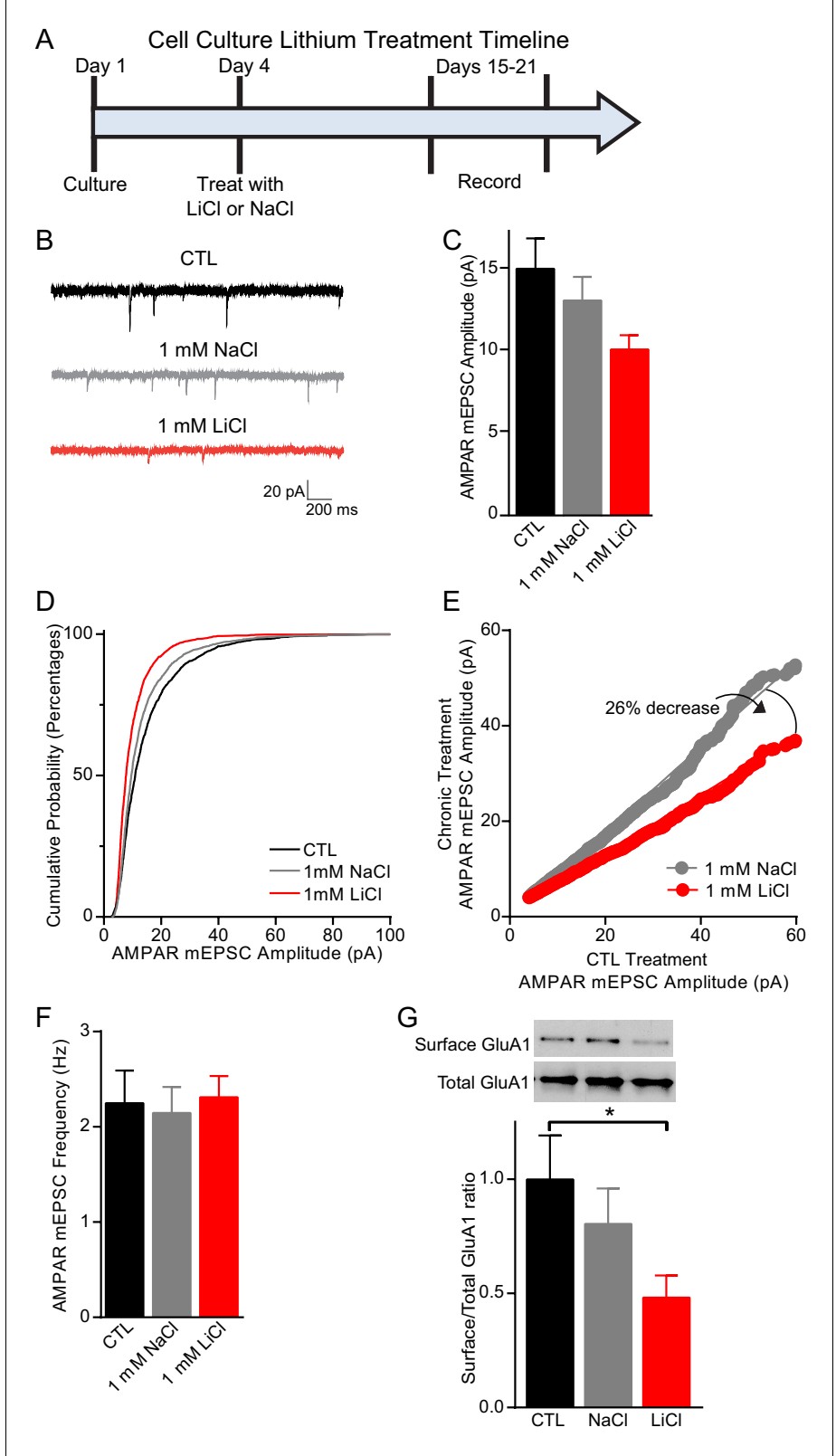

**Figure 3.** Chronic lithium treatment causes a significant decrease in synaptic scaling and surface GluA1 expression. (**A**) Timeline for treatment of dissociated hippocampal neurons with LiCl and NaCl. (**B**) Example traces of AMPAR mEPSCs from CTL untreated (top), 1 mM NaCl treated (middle) and 1 mM LiCl treated (bottom) dissociated hippocampal neurons. (**C**) Chronic LiCl treatment of cultured hippocampal neurons caused a trend

*Figure 3 continued on next page*

*Figure 3 continued*

towards decreased AMPA mEPSC average amplitudes in comparison to CTL neurons. Chronic NaCl treatment did not cause a change in average AMPA mEPSC amplitude in comparison to CTL neurons (ANOVA $F_{2,39} = 2.597$ p=0.0873, Bonferroni's multiple comparisons CTL vs NaCl p>0.999, CTL vs LiCl p=0.091). (**D**) Cumulative probability histogram showing a significant leftward shift (decrease) in the amplitudes of AMPAR-mEPSCs from cells chronically treated with 1 mM LiCl in comparison to CTL untreated and 1 mM NaCl treated neurons (Kolmogorov-Smirnov test: CTL vs 1 mM >LiCl *p=$1.75\times10^{-43}$, $D = 0.234$, 1 mM NaCl vs 1 mM LiCl *p=$8.14\times10^{-27}$, $D = 0.167$, $n$= 12–18 recordings per condition). (**E**) Rank order plot of CTL untreated AMPAR mEPSC amplitudes vs 1 mM LiCl AMPAR mEPSC amplitudes revealed a 41% decrease. Rank order plot of 1 mM NaCl AMPAR mEPSC amplitudes vs 1 mM LiCl AMPAR mEPSC amplitudes showed a 26% decrease. (CTL vs LiCl line of best fit y = 0.59x, CTL vs NaCl line of best fit y = 0.85x, Difference between NaCl and LiCl. 26, $n$ = 12–18 recordings per condition). (**F**) AMPAR-mEPSC frequency is indistinguishable between CTL untreated, NaCl treated, and LiCl treated neurons (ANOVA $F_{4,57} = 0.129$ p=0.97, $n$ = 12–18 recordings). (**G**) Surface biotinylation experiments revealed that chronic lithium treatment of hippocampal neurons results in a significant decrease in the surface/total GluA1 ratio (ANOVA $F_{2,21} = 2.911$ p=0.0766, Dunnett's multiple comparisons CTL vs NaCl p=0.58, CTL vs LiCl *p=0.04, $n$= 3 separate experiments).

BDNF expression. We investigated the effect of chronic lithium on synaptic function and found a significant synaptic downscaling of AMPAR mEPSC amplitudes that was dependent on BDNF as well as its high affinity receptor, TrkB. The decrease in AMPAR mEPSC amplitudes was due to a decrease in postsynaptic surface expression of GluA1 driven by dynamin-dependent AMPAR endocytosis, which could be acutely countered by administration of a dynamin inhibitor. Collectively, these data demonstrate that lithium can target AMPA receptor endocytosis to alter synaptic function.

In these experiments we used lithium treated chow to administer the drug to mice to more closely mimic the oral administration used by patients with bipolar disorder. In initial experiments, we established a chronic lithium regimen that resulted in clinically relevant lithium levels in serum. In each experiment we confirmed that an animal had a serum lithium level of 0.5–3.0 mM/L to ensure that the molecular and behavioral studies were performed under conditions similar to long-term treatment for Bipolar Disorder. We show that chronic lithium treatment in rodents causes a sustained increase in BDNF mRNA and protein in the hippocampus. However, BDNF was not required for the antidepressant action of lithium. Previous studies have shown that BDNF is required for the antidepressant effects of traditional antidepressants (*Adachi et al., 2008*; *Monteggia et al., 2004*), as well as rapid antidepressant effects of ketamine (*Autry et al., 2011*). While the current data showing lithium can exert an antidepressant effect in inducible *Bdnf* KOs is unexpected, the finding could be due to several possibilities. First, these data may suggest that mood stabilizers mediate an antidepressant response in a manner distinct from conventional antidepressants that target the monoamine system and independent of BDNF. Second, previous work with traditional antidepressants has largely focused on changes in *Bdnf* mRNA, with changes in BDNF protein not as fully explored leaving it unclear the time frame that conventional antidepressants increase BDNF protein (*Duman and Monteggia, 2006*). The rapid antidepressant action of ketamine requires BDNF and is protein translation dependent, with BDNF protein rapidly increased but then returning to baseline (*Autry et al., 2011*). In contrast, data from the current study shows that lithium treatment results in a sustained increase in *Bdnf* mRNA and protein expression. Taken together, these may data suggest acute manipulation of BDNF protein expression is involved in rapid antidepressant effects while a more chronic increase in BDNF expression is required for antimanic effects.

Lithium produces significant effects on synaptic transmission that may underlie its antimanic properties. Previous work has consistently shown that lithium application significantly decreases AMPAR mEPSC amplitudes in cultured neurons (*Ankolekar and Sikdar, 2015*; *Du et al., 2004*; *Gray et al., 2003*; *Seelan et al., 2008*; *Wei et al., 2010*). We replicated these data and showed chronic lithium treatment of hippocampal neurons results in a significant decrease in AMPA mEPSC amplitudes compared to control conditions. We also found that lithium treatment did not alter AMPA mEPSC frequency demonstrating specificity to the impact of the drug on synaptic transmission. Further analysis revealed that lithium produced a significant downward multiplicative synaptic scaling of all AMPAR-mEPSC amplitudes consistent with an impact on post-synaptic homeostatic plasticity. Previous in vivo and in vitro studies have shown that lithium decreases GluA1 and GluA2 surface

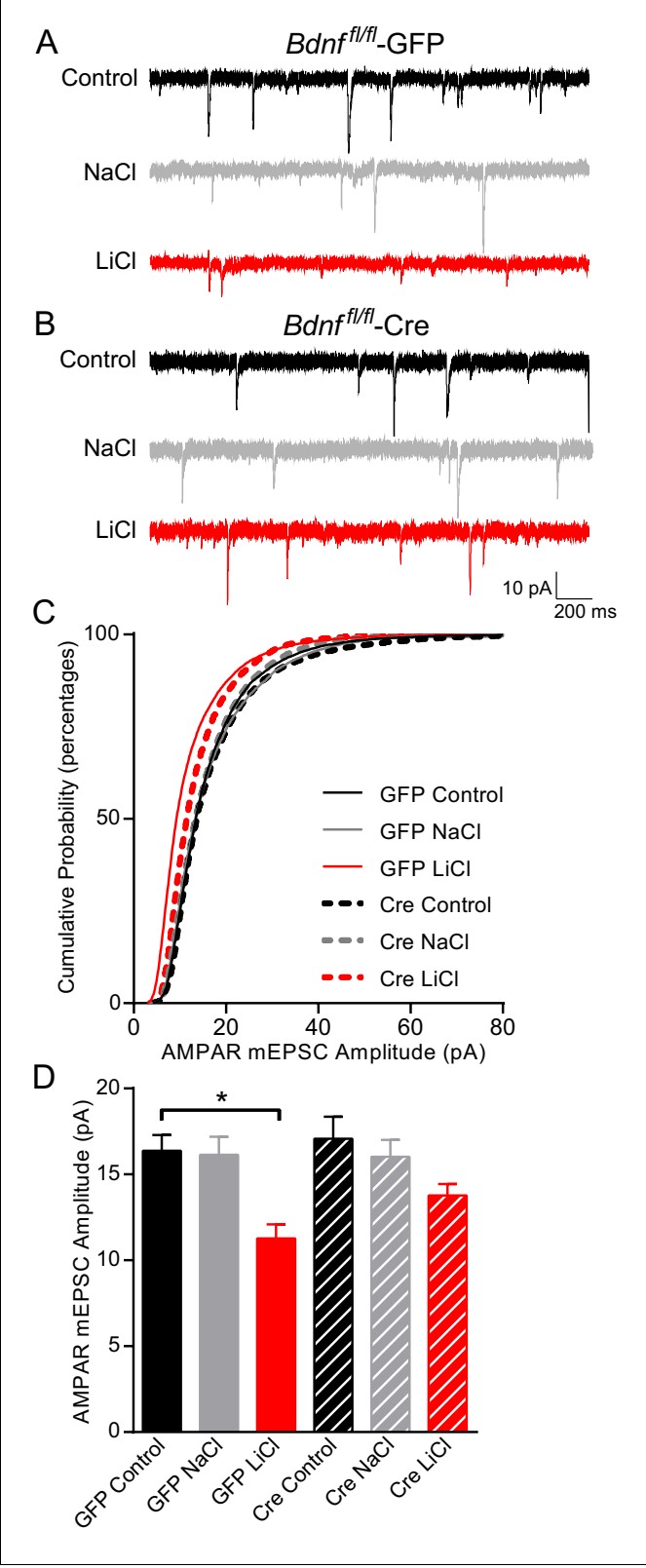

**Figure 4.** BDNF is required for the lithium-mediated decrease in AMPAR mEPSC amplitude. (**A**) Example traces from GFP infected *Bdnf* $^{fl/fl}$ neurons with no treatment (CTL, top) or 11–17 day incubation with 1 mM NaCl (middle) or 1 mM LiCl (bottom). (**B**) Example traces from Cre infected *Bdnf* $^{fl/fl}$ neurons with no treatment (control, top) or 11–17 day incubation with 1 mM NaCl (middle) or 1 mM LiCl (bottom). (**C**) Cumulative probability histogram

*Figure 4 continued on next page*

*Figure 4 continued*

showing a significant leftward shift (decrease) in AMPAR mEPSC amplitudes between *Bdnf* [fl/fl] -GFP control untreated and *Bdnf* [fl/fl] -GFP LiCl conditions. There is also a significant leftward shift (decrease) between *Bdnf* [fl/fl] -GFP LiCl and *Bdnf* [fl/fl] -Cre LiCl conditions (Kolmogorov-Smirnov test: GFP Control vs GFP LiCl p<0.0001 $D$ = 0.309, GFP LiCl vs Cre LiCl *p<0.0001 $D$ = 0.19, $n$ = 10–13 recordings per condition). (**D**) Lithium caused a significant decrease in AMPAR mEPSC amplitudes in *Bdnf* [fl/fl] -GFP neurons compared to control *Bdnf* [fl/fl] -GFP neurons. However, lithium did not impact AMPAR mEPSC amplitudes between *Bdnf* [fl/fl] -Cre neurons in comparison to untreated control *Bdnf* [fl/fl] -Cre neurons (ANOVA $F_{5,59}$ = 5.694 *p=0.0002, Tukey's multiple comparisons GFP Control vs GFP LiCl *p=0.003, Cre Control vs Cre LiCl p=0.197, $n$ = 10–13 recordings per condition).

expression in the hippocampus and in cultured cortical neurons (*Du et al., 2003*, *2008*; *Wei et al., 2010*). In addition, we found that in vivo lithium treatment caused a significant decrease in I/O curves in hippocampal slices, which is reminiscent of the decreased AMPA/NMDA ratio previously seen in rats following chronic lithium treatment (*Du et al., 2008*). Lithium treatment has also been shown to decrease phosphorylation of Thr840 on GluA1, which is associated with decreased AMPAR signaling (*Szabo et al., 2009*). In the current study, we found that lithium treatment of hippocampal neurons elicits synaptic downscaling due to decreased GluA1 surface expression demonstrating a previously unknown effect of lithium on post-synaptic homeostatic plasticity.

The lithium triggered significant decrease on AMPAR mEPSC amplitudes and hippocampal I/O curves was occluded in neurons lacking *Bdnf* suggesting a requirement in lithium's action. To confirm these data, we examined neurons with a deletion of *Ntrk2* and found that lithium's effect on AMPAR-mEPSC amplitudes was also occluded. While these data demonstrate that the effect of lithium on AMPA mEPSC amplitudes is dependent on BDNF and TrkB, the requirement for TrkB was more robust than BDNF per se. These findings may be due to low level expression of BDNF remaining after viral infection of Cre recombinase that acts on TrkB receptors or alternatively some minor effects of neurotrophin 3 (NT3) or neurotrophin 4 (NT4) activating the TrkB receptor to impact lithium's effects on synaptic transmission. Regardless, these results demonstrate a rather unexpected requirement for BDNF-TrkB signaling in lithium mediated effects on synaptic scaling as increased BDNF expression is typically associated with increased AMPAR surface expression (*Jakawich et al., 2010*; *Nakata and Nakamura, 2007*; *Nosyreva et al., 2013*). However, there is precedent for BDNF to induce downscaling of AMPAR surface expression. Specifically, acute increases in BDNF have been shown to increase GluA1 and GluA2 surface expression, whereas chronic increases in BDNF decrease surface expression of GluA1 and GluA2 (*Reimers et al., 2014*). Collectively, these data suggest that chronic lithium treatment through a sustained increase in BDNF expression leads to synaptic downscaling.

The BDNF-TrkB dependence of lithium-mediated effects on mEPSC amplitudes was due to decreased surface expression of GluA1, suggesting the removal of AMPARs by endocytosis. In neuronal synapses, dynamin has been shown to be a key mediator of endocytosis. To determine whether lithium promotes AMPAR endocytosis by a dynamin-dependent mechanism, we utilized a genetic approach with deletion of *Dnm1* and *Dnm3* and reversed the lithium-mediated decrease in AMPAR mEPSC amplitudes. To further establish that lithium triggers AMPAR endocytosis, we treated individual cells acutely with the dynamin inhibitor, Dyngo and demonstrated that lithium's impact on AMPAR endocytosis can be rapidly countered by dynamin inhibition. Previous work has shown that AMPAR internalization/endocytic membrane trafficking of the postsynaptic neuron can be regulated in a dynamin-dependent process to control strength (*Lu et al., 2007*). Taken together, our data show that lithium regulates dynamin-dependent endocytosis and rapidly removes AMPAR from the postsynaptic membrane altering synaptic efficacy.

Altered glutamatergic signaling and hyperactivity have been implicated in the pathology of bipolar disorder. Mania in bipolar patients has been associated with elevated glutamate signaling (*Lan et al., 2009*; *Ongür et al., 2008*). In post-mortem brain tissue of individuals with bipolar disorder an increase in expression of the vesicular glutamate transporter (VGlut1) mRNA and glutamate have been observed (*Eastwood and Harrison, 2010*; *Hashimoto et al., 2007*). Human induced pluripotent stem cell (IPSC) derived neurons from patients with bipolar disorder were shown to have

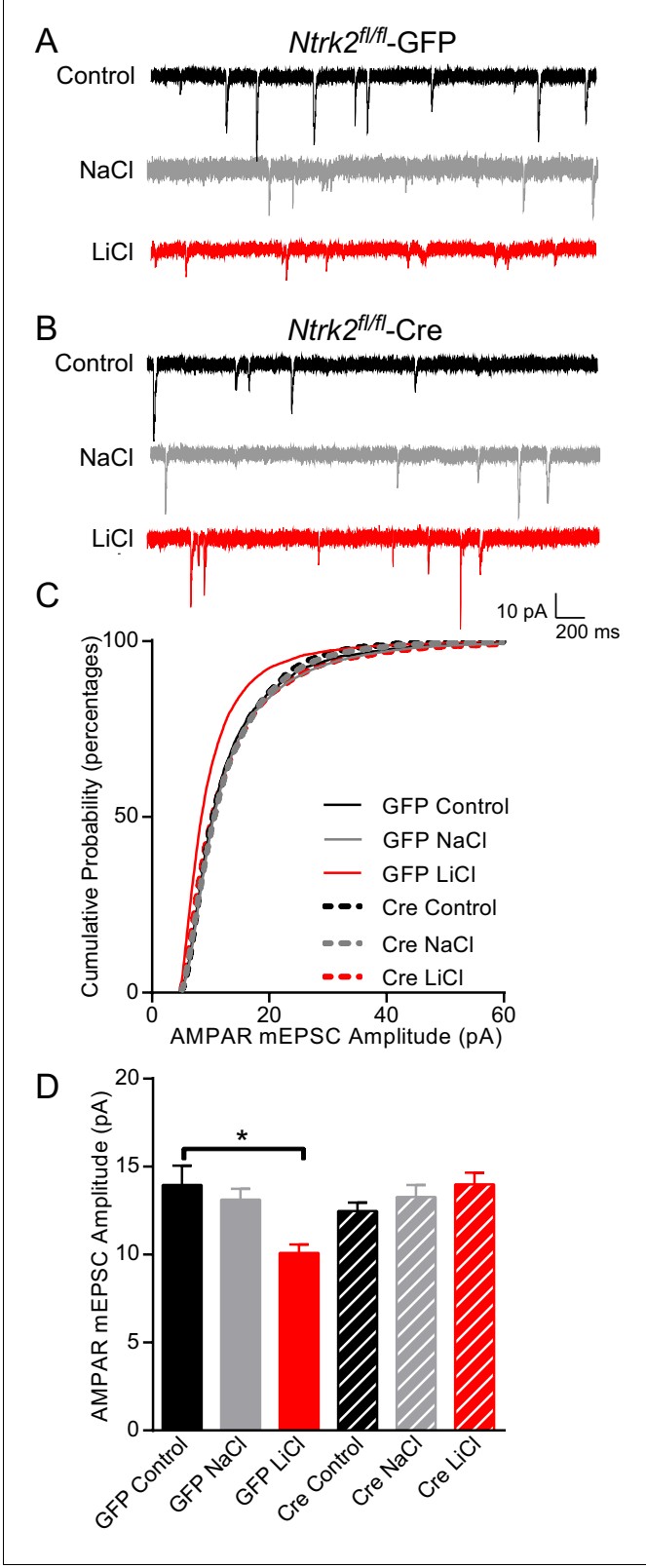

**Figure 5.** Lithium-mediated decrease in AMPAR mEPSC amplitude is dependent on TrkB. (**A**) Example traces from GFP infected *Ntrk2* $^{fl/fl}$ neurons, untreated Control (top), 1 mM NaCl treatment (middle), and 1 mM LiCl treatment (bottom). (**B**) Examples from Cre infected *Ntrk2* $^{fl/fl}$ neurons, untreated Control (top), 1 mM NaCl treatment (middle) and 1 mM LiCl treatment (bottom). (**C**) Cumulative probability histogram showing a significant leftward

*Figure 5 continued on next page*

*Figure 5 continued*

shift (decrease) in AMPAR-mEPSC amplitudes from *Ntrk2* $^{fl/fl}$ -GFP neurons treated with 1 mM LiCl. (Kolmogorov-Smirnov test: GFP control vs GFP LiCl *p<0.0001, $D$ = 0.181, GFP LiCl vs Cre LiCl *p<0.0001, $D$ = 0.153, $n$ = 9–17 recordings per condition). (D) Lithium caused a significant decrease in AMPAR-mEPSC amplitudes in *Ntrk2* $^{fl/fl}$ -GFP neurons compared to untreated control *Ntrk2* $^{fl/fl}$ -GFP neurons. However, lithium was unable to cause any significant changes in AMPAR-mEPSCs in *Ntrk2* $^{fl/fl}$ -Cre neurons compared to untreated *Ntrk2* $^{fl/fl}$ -Cre control (ANOVA $F_{5,76}$ = 5.107 *p=0.0004, Bonferonni multiple comparison test: GFP control vs GFP LiCl *p=0.002, Cre Control vs Cre LiCl p>0.999, $n$ = 9–17 recordings per condition).

lower action potential thresholds, increased evoked and spontaneous action potentials, and larger action potential amplitudes suggesting they are hyperactive in comparison to neurons derived from healthy controls (*Mertens et al., 2015*). Conversely, lithium treatment has been shown to reverse the glutamatergic hyperactivity from IPSC derived neurons from patients with bipolar disorder (*Mertens et al., 2015*). In addition, recent work has suggested that lithium may alter neuronal activity by regulating G-protein gated inward rectifier K+ (GIRK) channels (*Farhy Tselnicker et al., 2014*), which may modulate homeostatic scaling of AMPAR responses. Although, we cannot exclude lithium-mediated regulation of neuronal activity as a component of our observations, a parsimonious interpretation of our results suggests that direct biochemical effects of lithium mediated signaling is the key driver for the AMPAR downscaling we observed. Our findings showing that lithium triggers dynamin-dependent endocytosis of AMPARs may be a way in which to counteract the increased glutamatergic signaling associated with mania. Further work will be necessary to pursue this hypothesis and explore the role of lithium on dynamin-dependent endocytosis as a potential mechanism in antimanic action.

Findings from the current study identify a critical role for BDNF in the antimanic and synaptic effects of lithium treatment. These results demonstrate that lithium specifically impacts excitatory neurotransmission by decreasing AMPAR-mEPSCs through dynamin-dependent endocytosis to reduce surface expression of AMPAR subunits. These data start to provide a framework to elucidate the mechanisms underlying lithium's effect on synaptic transmission which may offer insight into novel therapeutic targets for the treatment of bipolar disorder.

## Materials and methods

### Animals

Male C57BL/6 mice aged 6–8 weeks were habituated to the animal facilities for at least 7 days prior to behavior testing. The mice were maintained on a 12 hr light/dark cycle with *ad libitum* access to food and water, unless otherwise noted for lithium treatment groups. The *Bdnf* $^{fl/fl}$ and *Ntrk2* $^{fl/fl}$ mice were generated as previously described and maintained as homozygous crosses (*Luikart et al., 2005*; *Rios et al., 2001*). The inducible *Bdnf* knockout (KO) mice were generated as previously described (*Monteggia et al., 2004*). The *Dnm1* $^{fl/fl}$ and the *Dnm3* $^{-/-}$ null KO mice were generated as previously described (*Ferguson et al., 2007*; *Raimondi et al., 2011*). All behavioral testing was done with age and weight matched male mice that were balanced by genotype. In these experiments we utilized only male mice as previous work from our laboratory has shown that loss of BDNF can impact depression-related behaviors in a sex-dependent manner (*Autry et al., 2009*). All electrophysiology and biochemistry experiments utilized male and female mice. All experiments were performed and data analyzed by an experimenter blind to drug condition and genotype. Animal protocols were approved by the Institutional Care and Use Committee at UT Southwestern Medical Center.

### Lithium treatment in vivo

The lithium treatment consisted of mice given 0.2% lithium chloride (LiCl) chow (Harlan Teklad, East Millstone, NJ) for 4 days and then switched to 0.4% LiCl (Harlan Teklad) chow for the remainder of the study, which lasted from 11 to 17 days total. All mice received water as well as a bottle of 0.9%

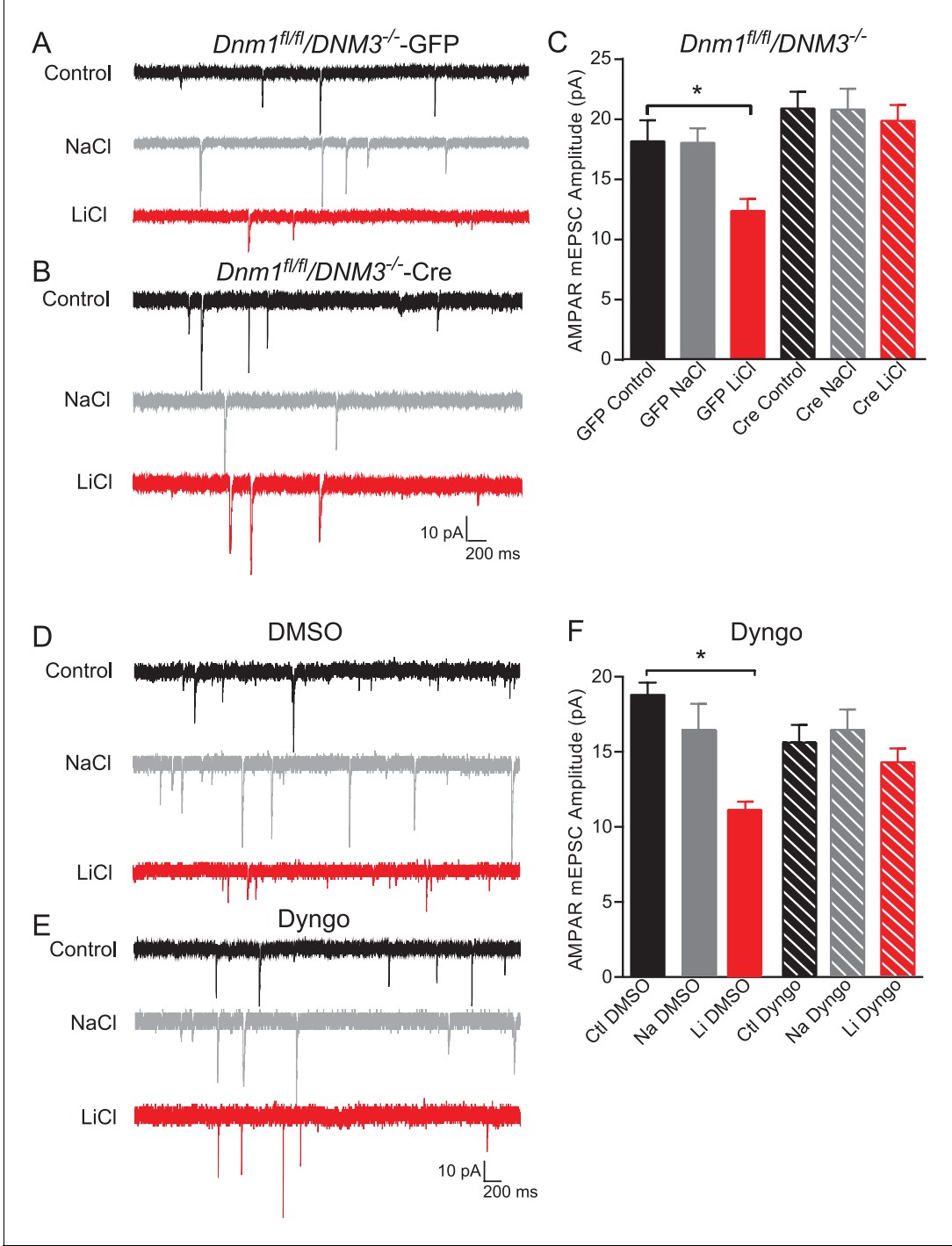

**Figure 6.** Dynamin-dependent endocytosis is required for lithium-mediated decrease in AMPAR-mEPSC amplitudes. (**A**) Example traces from GFP infected *Dnm1 $^{fl/fl}$/Dnm3$^{-/-}$* neurons, control (top), chronic NaCl treatment (1 mM, middle) and chronic LiCl treatment (1 mM, bottom). (**B**) Example traces from Cre infected *Dnm1 $^{fl/fl}$/Dnm3$^{-/-}$* neurons, control (top), chronic NaCl treatment (1 mM, middle), and chronic LiCl treatment (1 mM, bottom). (**C**) Chronic lithium treatment caused a significant decrease in AMPAR mEPSC amplitude in *Dnm1 $^{fl/fl}$/Dnm3$^{-/-}$* neurons infected with lenti-GFP virus compared to untreated *Dnm1 $^{fl/fl}$/Dnm3$^{-/-}$* neurons. However, there was no significant difference in AMPAR-mEPSC amplitudes between untreated and lithium treated *Dnm1 $^{fl/fl}$/Dnm3$^{-/-}$* neurons infected with lenti-CreGFP virus (ANOVA $F_{5,68}$ = 5.048 *p=0.0006, Bonferonni multiple comparison test: GFP control vs GFP LiCl *p=0.02, Cre Control vs Cre LiCl p=0.455, *n* = 8–12 recordings per condition). (**D**) Example traces from wildtype neurons recorded with DMSO-internal solution, control (top), chronic NaCl treatment (1 mM, middle), and chronic LiCl treatment (1 mM, bottom). (**E**) Example traces from wildtype neurons recorded with Dyngo-internal solution, control (top), chronic NaCl treatment (1 mM, middle), and chronic LiCl treatment (1 mM, bottom). (**F**) In comparison to control untreated neurons, chronic lithium treatment caused a significant decrease in AMPAR-mEPSC amplitudes. In

*Figure 6 continued on next page*

*Figure 6 continued*

contrast, lithium did not cause a significant change in AMPAR-mEPSC amplitudes in comparison to the untreated control when Dyngo is included in the internal pipette solution (ANOVA $F_{5,69}$ = 6.985, *p<0.0001, Bonferonni multiple comparison test: DMSO control vs DMSO LiCl *p<0.0001, Dyngo Control vs Dyngo LiCl p>0.999, $n$ = 11–17 recordings per condition).

sodium chloride (NaCl) to control for ion imbalances known to occur with lithium administration. Control mice were kept on identical chow except it did not contain lithium.

## Measurement of lithium in serum

Trunk blood was collected from all animals after the completion of the lithium treatment and behavioral testing. Whole blood was kept on ice until it was spun at 3000 RPM at 4°C for 10 min to separate red blood cells and serum. Lithium ion counts were made using a flame photometer (Jenway PFP7) and concentration was calculated following the construction of a standard curve. Mice with serum lithium concentrations below 0.5 mM and above 3 mM were excluded from behavioral and biochemical analyses.

## Behavior
### Forced swim test

The forced swim test (FST) was performed in accordance with published protocols (*Gideons et al., 2014*). In brief, mice were placed in a glass 4 L beaker with 3 L of 23 ± 2°C water for 6 min and test session were video-recorded. The last 5 min of each trial were scored by an observer blinded to drug condition and genotype to determine immobility time.

### Locomotor activity

Mice were habituated to the testing room for 1 hr and then given an intraperitoneal (i.p.) injection of amphetamine hydrochloride (Sigma, St. Louis, MO) at 2 mg/kg dissolved in saline or saline as a control. Mice were then immediately placed in standard home cages under red light and locomotor activity was measured for 2 hr by photocell beams linked to computer acquisition software (San Diego Instruments, San Diego, CA). The total beam counts for the 2 hr period were collected as a measurement of amphetamine induced hyperactivity (*Yates et al., 2007*).

## Hippocampal slice preparation and field potential electrophysiology

Mice chronically treated with lithium (11–17 days) were anesthetized with isoflurane before decapitation. The brain was removed and immersed for 2–3 min in ice-cold dissection buffer containing the following (in mM): 2.6 KCl, 1.25 $NaH_2PO_4$, 26 $NaHCO_3$, 0.5 $CaCl_2$, 5 $MgCl_2$, 212 sucrose and 10 glucose. The hippocampus was dissected out and cut with a vibratome into 400-μm-thick transverse sections in ice-cold dissection buffer continuously supplied with 95% $O_2$ and 5% $CO_2$. Area CA3 was surgically removed from each slice immediately after sectioning. The sections recovered in oxygenated ACSF containing the following (in mM): 124 NaCl, 5KCl, 1.25 $NaH_2PO_4$, 26 $NaHCO_3$, 2 $CaCl_2$, 2 $MgCl_2$ and 10 glucose, pH 7.4 (continuously equilibrated with 95% $O_2$ and 5% $CO_2$) for 2–3 hr at 30°C. Hippocampal slices were transferred to the recording chamber and perfused with ACSF at a rate of 2–3 ml/min at 30°C. Field EPSPs (fEPSPs) were evoked by inserting a bipolar platinum-tungsten stimulating electrode (Frederick Haer, Bowdoin, ME) into Schaffer collateral/commissural afferents. The extracellular recording electrodes filled with ACSF (resistance, 1–2 MΩ) were placed into the CA1 just beneath the molecular layer. Baseline responses were collected every 30 s using an input stimulus intensity that induced 30–40% of the maximum response. After a 20 min stable baseline, an ascending series of stimulus input intensities (range, 5 to 200 μA) were applied and the input/output (I/O) curve was plotted by the slope of fEPSP versus afferent volley amplitude. Data are presented as mean ± SEM and analyzed using two-way ANOVA. Sidak's multiple comparisons test were conducted after significant interaction effects were found. A p value of <0.05 was required for statistical significance.

## Cell culture

Dissociated hippocampal cultures were prepared as previously described (*Kavalali et al., 1999*; *Reese and Kavalali, 2015*). Whole hippocampi were dissected from postnatal day 0–3 (P0-P3) mice. The hippocampi were trypsinized (~10 mg/mL trypsin; Invitrogen), dissociated mechanically, and plated on Matrigel (Corning Biosciences, Tewksbury, MA)-coated coverslips for electrophysiology or directly onto tissue culture treated plates for protein collection for western blot analysis. At 1 d in vitro (DIV), 4 µM cytosine arabinoside (ARAC; Sigma) was added. At 4 DIV, the ARAC concentration was decreased to 2 µM with a media change. Treatment with LiCl (1 mM) solution (Sigma) or NaCl (1 mM) solution (Sigma) was initiated at 4 DIV and lasted for 11–17 days. The $Bdnf^{fl/fl}$, $Ntrk2^{fl/fl}$, and $Dnm1^{fl/fl}$ / $Dnm3^{-/-}$ cultures were infected with lentivirus expressing Cre recombinase (Cre) tagged with GFP or lentivirus expressing GFP alone as a control at 4 DIV. Lentivirus constructs and virus preparation from HEK293T/17 cells were prepared as previously described (*Akhtar et al., 2009*). HEK293T/17 cell line was purchased from ATCC (Cat. Number: CRL-11268). All electrophysiology experiments and protein collected for western blot analysis were done on 15–21 DIV cultures.

## Patch-clamp electrophysiology

Whole-cell patch clamp recordings were performed on hippocampal pyramidal neurons as previously described (*Gideons et al., 2014*). The external Tyrode's solution contained (in mM): 150 NaCl, 4 KCl, 2 CaCl$_2$, 1.25 MgCl$_2$, 10 glucose, and 10 Hepes (ph 7.4) at ~300 mOsm. The pipette internal solution contained (in mM): 110 K-gluconate, 20 KCl, 10 NaCl, 10 Hepes, 0.6 EGTA, 4 Mg-ATP, 0.3 Na-GTP, and 10 lidocaine N-ethyl bromide (pH 7.3) at ~300 mOsm. Pipettes had a resistance between 3–7 MΩ. The junction potential between the internal and external solutions was ~12 mV and was subtracted from all recordings. AMPAR-mediated mEPSCs were recorded in the presence of 50 µM (2R)-amino-5-phosphonovaleric acid (AP5; Tocris, Bristol, UK), 1 µM tetrodotoxin (TTX; EMD Millipore, Billerica, MA), and 50 µM picrotoxin (PTX; Sigma). For the acute dynamin inhibitor studies, dynamin activity was inhibited in the postsynaptic neuron by adding 30 µM Dyngo-4A (Abcam, Cambridge, MA) to the internal pipette solution (*McCluskey et al., 2013*). DMSO was added to the internal pipette solution as a control for these experiments. Data were acquired using a MultiClamp 700B amplifier and Clampex 10.0 software (Molecular Devices, Sunnyvale, CA). Recordings were sampled at 100 µs, filtered at 2 kHz with a gain of 5. No more than three recordings were obtained per coverslip. AMPAR-mEPSCs were analyzed from a 3–5 min recording using MiniAnalysis software by an experimenter blind to drug condition and genotype.

## Quantitative RT-PCR

Briefly, fresh frozen hippocampi were dissected and total RNA was extracted using Trizol (Invitrogen) following the manufacturer's instruction. Conditions for cDNA synthesis, amplification, and primer sequences were previously described (*Adachi et al., 2008*).

## Protein quantification

Anterior hippocampal slices (~1 mm thick, 2–3 per mouse) were dissected and flash-frozen following 11 days of lithium treatment or immediately following the last behavioral test depending on the experiment. Hippocampal tissue was lysed in a radio immunoprecipitation assay (RIPA) buffer containing: 50 mM Tris pH 7.4, 1% Igepal, 0.1% SDS, 0.5% Na deoxycholate, 4 mM EDTA, 150 mM NaCl, phosphatase inhibitors (10 mM Na pyrophosphate, 50 mM NaF, 2 mM Na orthovanadate), and protease inhibitors (cOmplete Mini tablets, Roche, Basel, Switzerland). Protein concentration was quantified with the Quick-Start Bradford assay (Bio-Rad, Hercules, CA). Approximately 30 µg of protein was electrophoresed on SDS-PAGE gels and then transferred to nitrocellulose membranes. The membranes were incubated with primary antibodies overnight at the following dilutions: BDNF (Abcam, Cat. Number: ab108319; RRID: AB_10862052), 1:1000, GAPDH (Cell Signaling, Cat. Number: 2118S; RRID: AB_561053) 1:50,000, phosphorylated GSK3$\beta$ (Ser9, Cell Signaling, Cat. Number: 9323S) and total GSK3$\beta$ (Cell Signaling, Cat. Number: 9315S) 1:30,000. Primary antibodies for phospho-GSK3$\beta$ included 5% BSA. After washing, the membranes were incubated in anti-rabbit secondary antibodies: BDNF, 1:5000, GAPDH, 1:10,000, phospho-GSK3$\beta$ and total GSK3$\beta$, 1:10,000. Protein bands were detected using ECL then exposed to film. Following development of phospho-GSK3$\beta$, membranes were stripped using Restore PLUS Western Blot Stripping Buffer

(ThermoScientific, Waltham, MA), put in block, and then in primary antibody for total GSK3$\beta$. BDNF expression was normalized to GAPDH while a ratio of phospho- GSK3$\beta$ intensity to total GSK3$\beta$ was examined. The phospho-GSK3$\beta$ and total GSK3$\beta$ antibodies are known to recognize doublet bands (Cell Signaling).

### Cell surface AMPAR expression measurement

Membrane biotinylation experiments were performed as previously described (*Nosyreva et al., 2013*). Dissociated hippocampal cultures from C57BL/6 mice were incubated in Tyrode's solution containing 1 mg/ml sulfo-NHS-LC-biotin (Pierce/Thermofisher) for 20 min on ice. The biotin reactions were quenched by incubating the cultures in Tris-buffered saline (TBS) with 15 mM ammonium chloride for 5 min on ice, and then washed twice with TBS for 5 min on ice. Following the second TBS wash, the cultures were lysed in RIPA buffer (as described above) for 10 min on ice and spun at 12,000 rpm for 5 min to remove non-solubilized material. Total protein concentration was quantified by Quik-Start Bradford assay (Bio-rad). 100 µg of protein from each sample was incubated with 100 µL of washed UltraLink NeutrAvidin (Pierce/Thermofisher) immobilized beads and rotated overnight at 4°C. Beads were washed with three times with RIPA buffer, followed by three washes with TBS at 4°C. Protein was eluted from the beads with SDS-PAGE sample buffer supplemented with $\beta$-mercaptaethanol (BME) for 10 min at 95°C. Eluted surface protein and 20 µg of total protein in SDS-PAGE-BME sample buffer were resolved by 10% SDS-PAGE gel, transferred to nitrocellulose, and probed with anti-GluA1 antibody (1:1000, Chemicon/EMD Millipore, Cat. Number: MAB2263) and anti-GAPDH antibody (1:50,000) overnight. Secondary anti-rabbit antibodies were at 1:2000 and 1:10,000 for GluA1 and GAPDH respectively. Surface GluA1 over total GluA1 ratio is reported.

### Statistics

Data are reported as mean ± SEM. Statistical differences in the FST, locomotor tests, western blot, QPCR, mEPSC amplitude and frequency were assessed using unpaired two-tailed Student's *t* test or one-way ANOVA when appropriate. Tukey and Bonferroni *post hoc* tests were used when appropriate. Differences in the cumulative probability histograms were assessed with Kolmogorov-Smirnov test. Statistical significance was defined at p≤0.05. Statistical outliers were identified with the Robust regression and Outlier removal (ROUT) method. All t-test, ANOVAs, associated post hoc tests, and ROUT method statistical tests were performed with Prism 6 (GraphPad, La Jolla, California). The Kolmogorov-Smirnov test was performed using Past 3.02 (http://folk.uio.no/ohammer/past/).

## Acknowledgements

We thank members of the Monteggia laboratory for helpful advice and discussions on the manuscript. We thank Dr. Rudolf Jaenisch for the *Bdnf* $^{fl/fl}$ mice and Dr. Luis Parada for the *NtrkB* $^{fl/fl}$ mice. This work was supported by National Institutes of Health Grants MH070727 (to LMM) and MH066198 (to ETK) as well as awards from the Brain and Behavior Research Foundation and the International Mental Health Research Organization (to LMM).

## Additional information

### Funding

| Funder | Grant reference number | Author |
| --- | --- | --- |
| National Institute of Mental Health | MH070727 MH066198 | Ege T Kavalali<br>Lisa M Monteggia |
| Brain and Behavior Research Foundation | Distinguished Investigator Award | Ege T Kavalali<br>Lisa M Monteggia |
| International Mental Health Research Organization | Research Award | Lisa M Monteggia |

The funders had no role in study design, data collection and interpretation, or the decision to submit the work for publication.

## Author contributions
ESG, Data curation, Formal analysis, Investigation, Methodology, Writing—original draft; P-YL, MM, Data curation, Formal analysis, Methodology; ETK, Conceptualization, Formal analysis, Funding acquisition, Methodology, Writing—review and editing; LMM, Conceptualization, Funding acquisition, Methodology, Writing—review and editing

## Author ORCIDs
Erinn S Gideons, http://orcid.org/0000-0003-4778-3869
Pei-Yi Lin, http://orcid.org/0000-0001-7447-3212
Melissa Mahgoub, http://orcid.org/0000-0003-1491-4439
Ege T Kavalali, http://orcid.org/0000-0003-1777-227X
Lisa M Monteggia, http://orcid.org/0000-0003-0018-501X

## Ethics
Animal experimentation: Animal protocols were approved by the Institutional Care and Use Committee at UT Southwestern Medical Center (UTSW APN# 2017-101831G).

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
