## [Decision Letter]

Thank you for submitting your article "Chronic lithium treatment elicits its antimanic effects via BDNF-TrkB dependent synaptic downscaling" for consideration by *eLife*. Your article has been reviewed by three peer reviewers, one of whom is a member of our Board of Reviewing Editors, and the evaluation has been overseen by a Senior Editor. The following individual involved in review of your submission has agreed to reveal their identity: Lu Chen (Reviewer #3).

The reviewers have discussed the reviews with one another and the Reviewing Editor has drafted this decision to help you prepare a revised submission

Summary:

This is a relatively straightforward but highly significant study demonstrating that chronic lithium administration to mice reduces amphetamine-induced locomotion (a model for anti-manic effects) through a mechanism that depends on BDNF-TrkB signaling, whereas lithium's ability to reduce immobility in the forced swim test (a model for anti-depressant effects) does not depend on BDNF. It is particularly compelling that both tests were conducted in the same cohort of mice. Furthermore, patch-clamp recordings showed that lithium reduced AMPAR-mediated mEPSC amplitude in cultured hippocampal neurons in a manner reminiscent of homeostatic downscaling, and that this effect also required BDNF-TrkB pathway. The decrease in AMPAR-mediated mEPSC amplitude in cultured hippocampal neurons was associated with reduced surface expression of GluA1 subunits. Two different strategies for inhibiting dynamin-dependent endocytosis rescued the lithium-induced decrease in mEPSC amplitude. These are novel findings that will be of broad interest. Experiments are well designed and the effects are convincing.

Here are some suggestions for the authors to consider in order to strengthen their findings and clarify the conclusions:

Essential revisions:

1) It remains unclear what is the signal triggering mEPSC downscaling. Does lithium trigger an acute increase in spontaneous neuronal activity probably through inhibition of GIRK channels (Farhy Tselnicker et al., 2014)? If so, it would justify the 'homeostatic' nature of a negative feedback response through the proposed postsynaptic downscaling.

2) The authors should provide further insight into the downstream signaling mechanisms that lead to down-regulation of mEPSC amplitude by lithium. In particular, it is important to better understand whether inhibition of GSK-3beta is involved in BDNF-induced synaptic downscaling by lithium. It can be tested by using specific inhibitors of GSK-3beta and examining if they occlude lithium-induced synaptic downscaling. As Akt can be activated by numerous growth factors, including BDNF, it may down-regulate synaptic AMPARs by inhibiting GSK3beta.

3) All recordings are done in cultured neurons. Although others have apparently shown that in vivo lithium exposure reduces mEPSC amplitude (three studies are cited), were those recordings done in hippocampal neurons and under conditions similar to those studied here (e.g., lithium dosing? age of animal?). If not, it would be important to use slice physiology to confirm in the current mouse model that lithium reduces mEPSC amplitude and perhaps show that this is absent in slices from BDNF knockout mice.

4) Is a reduction in amphetamine-induced locomotor activity an accepted model for lithium's anti-manic effects? Even if so, the authors should be a little more conservative in referring to lithium's action in this test as "anti-manic".

---

## [Author Response]

*Essential revisions:*

*1) It remains unclear what is the signal triggering mEPSC downscaling. Does lithium trigger an acute increase in spontaneous neuronal activity probably through inhibition of GIRK channels (Farhy Tselnicker et al., 2014)? If so, it would justify the 'homeostatic' nature of a negative feedback response through the proposed postsynaptic downscaling.*

We thank the reviewer for this suggestion. We conducted additional experiments to test this premise (i.e. lithium triggering an acute increase in activity) and, if anything, in our system, we detect an acute decrease in activity (the frequency of spontaneous action potential firing in the absence of lithium 0.96 ± 0.27 Hz and the presence of lithium 0.22 ± 0.06 Hz in external solution) which is not consistent with the downscaling we observed. While, we cannot exclude lithium-mediated regulation of neuronal activity as a component of our observations, a parsimonious interpretation of our results suggest that direct biochemical effects of lithium mediated signaling is the key driver for the downscaling. We now include this point in the Discussion.

*2) The authors should provide further insight into the downstream signaling mechanisms that lead to down-regulation of mEPSC amplitude by lithium. In particular, it is important to better understand whether inhibition of GSK-3beta is involved in BDNF-induced synaptic downscaling by lithium. It can be tested by using specific inhibitors of GSK-3beta and examining if they occlude lithium-induced synaptic downscaling. As Akt can be activated by numerous growth factors, including BDNF, it may down-regulate synaptic AMPARs by inhibiting GSK3beta.*

To address this comment we have performed additional experiments in which we examined an acute 24-hour treatment of the GSK-3beta inhibitor SB216763 on C57BL/6 hippocampal neurons and did not observe an effect on mEPSC amplitude (DMSO Control: 16.06 ± 1.346 S.E.M. vs SB21673: 16 ± 1.178 S.E.M., *n*=10-12 recordings per condition). The lack of an effect of the GSK-3beta inhibitor on mEPSC amplitudes may be due to the acute nature of the treatment, therefore we chronically treated C57Bl/6 hippocampal neurons with SB216763 for 11-15 days as was done with lithium chloride in our previous experiments. However, chronic SB216763 treatment compromised the survival of cultured neurons, which made the interpretation of these experiments difficult.

*3) All recordings are done in cultured neurons. Although others have apparently shown that in vivo lithium exposure reduces mEPSC amplitude (three studies are cited), were those recordings done in hippocampal neurons and under conditions similar to those studied here (e.g., lithium dosing? age of animal?). If not, it would be important to use slice physiology to confirm in the current mouse model that lithium reduces mEPSC amplitude and perhaps show that this is absent in slices from BDNF knockout mice.*

We have followed the reviewers’ advice and tested the chronic in vivo impact of lithium on the CA3-CA1 synapses recorded from hippocampal slices. In this setting, we could detect a clear decrease in input/output curves obtained from animals chronically treated with lithium (Figure 2). Moreover, this decrease in input/output slope was not detected in mice lacking BDNF (Figure 2). We have also modified the text to make it more apparent that the previous studies showing lithium exposure reduces mEPSC amplitude were performed on hippocampal neurons in culture.

*4) Is a reduction in amphetamine-induced locomotor activity an accepted model for lithium's anti-manic effects? Even if so, the authors should be a little more conservative in referring to lithium's action in this test as "anti-manic".*

The amphetamine hyperactivity test is the most commonly used test assessing ‘anti-manic effects’ in rodents. Therefore, we used similar language in our original submission. Nevertheless, we agree with the reviewer regarding caution in interpreting lithium’s effects as anti-manic in rodent models. In the revised manuscript we have tried to dampen the interpretation of the in vivo behavioral data and now refer to lithium’s behavioral action as ‘antimanic-like’ in the study.